# A Combination of Metagenomic and Cultivation Approaches Reveals Hypermutator Phenotypes within *Vibrio cholerae*-Infected Patients

Inès Levade,[a,h] Ashraful I. Khan,[b] Fahima Chowdhury,[b] Stephen B. Calderwood,[c,d,e] Edward T. Ryan,[c,d] Jason B. Harris,[c,f] Regina C. LaRocque,[c,d] [ID] Taufiqur R. Bhuiyan,[b] Firdausi Qadri,[b] Ana A. Weil,[g] [ID] B. Jesse Shapiro[a,h,i]

aDepartment of Biological Sciences, University of Montreal, Montreal, Quebec, Canada
bCenter for Vaccine Sciences, International Centre for Diarrhoeal Disease Research, Dhaka, Bangladesh
cDivision of Infectious Diseases, Massachusetts General Hospital, Boston, Massachusetts, USA
dDepartment of Medicine, Harvard Medical School, Boston, Massachusetts, USA
eDepartment of Microbiology, Harvard Medical School, Boston, Massachusetts, USA
fDepartment of Pediatrics, Harvard Medical School, Boston, Massachusetts, USA
gDivision of Allergy and Infectious Diseases, University of Washington, Seattle, Washington, USA
hDepartment of Microbiology and Immunology, McGill University, Montreal, Quebec, Canada
iMcGill Genome Centre, Montreal, Quebec, Canada

**ABSTRACT** *Vibrio cholerae* can cause a range of symptoms, from severe diarrhea to asymptomatic infection. Previous studies using whole-genome sequencing (WGS) of multiple bacterial isolates per patient showed that *V. cholerae* can evolve modest genetic diversity during symptomatic infection. To further explore the extent of *V. cholerae* within-host diversity, we applied culture-based WGS and metagenomics to a cohort of both symptomatic and asymptomatic cholera patients from Bangladesh. While metagenomics allowed us to detect more mutations in symptomatic patients, WGS of cultured isolates was necessary to detect *V. cholerae* diversity in asymptomatic carriers, likely due to their low *V. cholerae* load. Using both metagenomics and isolate WGS, we report three lines of evidence that *V. cholerae* hypermutators evolve within patients. First, we identified nonsynonymous mutations in *V. cholerae* DNA repair genes in 5 out of 11 patient metagenomes sequenced with sufficient coverage of the *V. cholerae* genome and in 1 of 3 patients with isolate genomes sequenced. Second, these mutations in DNA repair genes tended to be accompanied by an excess of intrahost single nucleotide variants (iSNVs). Third, these iSNVs were enriched in transversion mutations, a known hallmark of hypermutator phenotypes. While hypermutators appeared to generate mostly selectively neutral mutations, nonmutators showed signs of convergent mutation across multiple patients, suggesting *V. cholerae* adaptation within hosts. Our results highlight the power and limitations of metagenomics combined with isolate sequencing to characterize within-patient diversity in acute *V. cholerae* infections, while providing evidence for hypermutator phenotypes within cholera patients.

**IMPORTANCE** Pathogen evolution within patients can impact phenotypes such as drug resistance and virulence, potentially affecting clinical outcomes. *V. cholerae* infection can result in life-threatening diarrheal disease or asymptomatic infection. Here, we describe whole-genome sequencing of *V. cholerae* isolates and culture-free metagenomic sequencing from stool of symptomatic cholera patients and asymptomatic carriers. Despite the typically short duration of cholera, we found evidence for adaptive mutations in the *V. cholerae* genome that occur independently and repeatedly within multiple symptomatic patients. We also identified *V. cholerae* hypermutator phenotypes within several patients, which appear to generate mainly neutral or deleterious mutations. Our work sets the stage for future

Address correspondence to B. Jesse Shapiro, jesse.shapiro@mcgill.ca.

Our results highlight the power and limitations of metagenomics combined with isolate sequencing to characterize within-patient diversity in acute V. cholerae infections, while providing evidence for hypermutator phenotypes within cholera patients.

studies of the role of hypermutators and within-patient evolution in explaining the variation from asymptomatic carriage to symptomatic cholera.

**KEYWORDS** *Vibrio cholerae*, cholera, metagenomics, within-patient evolution, hypermutation, asymptomatic carriage, convergent evolution, genomics, intrahost diversity, natural selection, population genetics

Infection with *Vibrio cholerae*, the etiological agent of cholera, causes a clinical spectrum of symptoms that range from asymptomatic colonization of the intestine to severe watery diarrhea that can lead to death. Although absent from most resource-rich countries, this severe diarrheal disease still plagues many developing nations. According to the WHO, there are an estimated 1.3 to 4.0 million cases of cholera each year, with 21,000 to 143,000 deaths worldwide (1). Cholera occurs predominantly in areas of endemicity but can also cause explosive outbreaks, as seen in Haiti in 2010 or in Yemen, where over 2.2 million cases are suspected since 2016 (2, 3). Although cholera vaccines have reduced disease in some areas, the increasing number of people lacking access to sanitation and safe drinking water, the emergence of a pandemic lineage of *V. cholerae* with increased virulence (4), and environmental persistence of this waterborne pathogen underscore the need to understand and interrupt transmission of this disease.

Cholera epidemiology and evolutionary dynamics have been studied by high-throughput sequencing technologies and new modeling approaches, at both global and local scales (5, 6). Yet, many questions remain regarding asymptomatic carriers of *V. cholerae*, including their role and importance in the transmission chain during an epidemic (7, 8). Numerous observational studies have identified host factors that could impact the severity of symptoms, including lack of preexisting immunity, blood group O status, age, polymorphisms in genes of the innate immune system, or variation in the gut microbiome (9–13).

Recent studies have shown that despite the acute nature of *V. cholerae* infection, which typically lasts only a few days, genetic diversity can appear and be detected in a *V. cholerae* population infecting individual patients (14, 15). In a previous study, we sampled multiple *V. cholerae* isolates from each of eight patients (five from Bangladesh and three from Haiti) and sequenced 122 bacterial genomes in total. Using stringent controls to guard against sequencing errors, we detected a few (0 to 3 per patient) within-patient intrahost single nucleotide variants (iSNVs) and a greater number of gene content variants (on the order of ~100 gene gain/loss events within patients) (15). This variation may affect adaptation to the host environment, either by resistance to phage predation (14) or by impacting biofilm formation (15).

Several pathogens are known to evolve within human hosts (16), and hypermutation has been observed in some cases due to loss-of-function mutations in the mismatch repair machinery (17–19). While these hypermutators may quickly acquire adaptive mutations, they also bear a burden of deleterious mutations (20). For the population to survive the burden of deleterious mutations, hypermutators may revert to a nonmutator state or may recombine their adaptive alleles into the genomes of nonmutators in the population (17, 21). The hypermutator phenotype has been observed in vibrios in the aquatic environment (22), and induced in *V. cholerae* in an experimental setting (23), but not clearly documented within infected patients. There is some evidence for hypermutation in *V. cholerae* clinical strains isolated between 1961 and 1965 (24); however, the authors recognized that these hypermutators could also have emerged during long-term culture (25). It therefore remains unclear if hypermutators readily evolve within individual cholera patients.

When within-patient pathogen populations are studied with culture-based methods, their diversity may be underestimated because the culture process can select isolates more suited to growth in culture and due to undersampling of rare variants (26). In this study, we used a combination of culture-free metagenomics and whole-genome

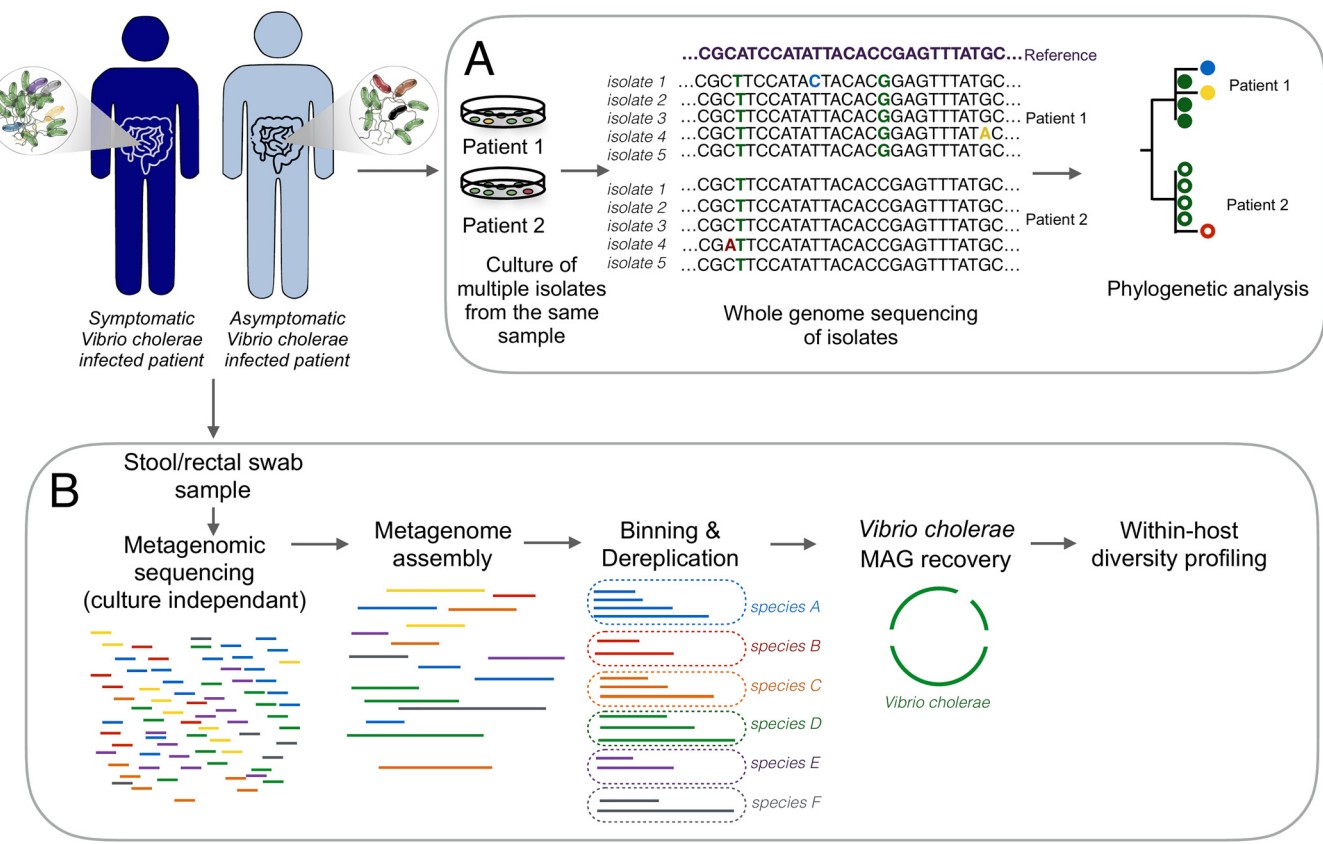

**FIG 1** Summary of the culture-dependent and culture-free metagenomics workflows for the characterization of *Vibrio cholerae* within-patient diversity. Stool or rectal swab samples were collected from symptomatic and asymptomatic *Vibrio cholerae*-infected individuals and processed using two different approaches: culture, DNA extraction, and whole-genome sequencing of multiple isolates per patient (A); and genome-resolved metagenomics involving DNA extraction directly from a microbiome sample, followed by sequencing, assembly, genome binning, and dereplication to generate metagenome-assembled genomes (MAGs), and within-host diversity profiling by mapping reads back to the MAGs (B).

sequencing (WGS) of a limited number of cultured isolates to characterize the within-patient diversity of *V. cholerae* in individuals with different clinical syndromes ranging from symptomatic to asymptomatic infection. Using both approaches, we report evidence of *V. cholerae* hypermutators within both symptomatic and asymptomatic infected patients. These hypermutators are characterized by a high mutation rate and accumulation of an excess of likely neutral or deleterious mutations in the genome. Finally, we provide evidence of adaptive mutations occurring in nonmutator *V. cholerae* infections.

## RESULTS

**Taxonomic analyses of metagenomics sequences from *Vibrio cholerae*-infected index cases and household contacts.** To evaluate the level of within-patient diversity of *Vibrio cholerae* populations infecting symptomatic and asymptomatic patients in a cohort in Dhaka, Bangladesh, we used both culture-based whole-genome sequencing and culture-free shotgun metagenomic approaches (Fig. 1). Cholera patients and their household contacts were enrolled because household contacts of cholera patients are known to be at high risk of *V. cholerae* infection during the week after an index case of cholera occurs in the household (11). Infected household contacts are known to exhibit a range of clinical outcomes from asymptomatic to severe symptomatic disease. We enrolled patients from February 2013 to May 2014, collected demographic information, and obtained rectal swabs, immunologic measures, and symptom histories during 30 days of prospective follow-up, as previously described (11–13). We excluded household contacts who reported antibiotic use during the week prior to enrollment.

We performed metagenomic sequencing of 22 samples from 21 index cases and 11 samples from 10 household contacts infected with *Vibrio cholerae*, of which two

remained asymptomatic during the 30-day follow-up period (see Table S1 in the supplemental material). After removal of reads mapping to the human genome, we used Kraken2 and MIDAS to taxonomically classify the remaining reads and identify samples with enough *Vibrio cholerae* reads to reconstruct genomes. Among symptomatic patients (index cases and household contacts), 15 samples from 14 patients contained enough reads to reconstruct the *Vibrio cholerae* genome with a mean depth of coverage of $>5\times$. Neither of the two asymptomatic patients had enough *Vibrio cholerae* reads in their metagenomic sequences to reconstruct genomes by mapping or *de novo* assembly (mean coverage depth, $<0.05\times$). We also detected reads from two *Vibrio* phages (ICP1 and ICP3) in some of these samples (Table S1).

**Recovery of high-quality *Vibrio cholerae* MAGs from metagenomic samples.** To reconstruct *Vibrio cholerae* metagenomic assembled genomes (MAGs) from the 11 samples with a coverage depth of $>10\times$, we *de novo* assembled each sample individually except that from patient E, for whom we coassembled two samples from two consecutive sampling days. High-quality MAGs identified as *Vibrio cholerae* were obtained from each assembly, with no redundancy and with completeness ranging from 91% to 100% (Table S2). We dereplicated the set of bins and removed all but the highest-quality genome from each redundant set, identifying the bin from patient J as the highest-quality MAG overall, which was used as a *V. cholerae* reference for read mapping and SNV calling.

**_Vibrio cholerae_ within-patient nucleotide diversity estimated from metagenomic data.** All metagenomes with a *Vibrio cholerae* mean coverage depth of $>5\times$ were mapped against the dereplicated genome set, and we assessed within-patient genetic diversity using InStrain (27). This program reconstructs the "strain cloud" of a bacterial population by mapping metagenomic reads to metagenome-assembled genomes (MAGs) and calculates the allele frequency of each single nucleotide variant (SNV) in the population. To remove potential false-positive SNVs (due to sequencing errors or mismapping of reads belonging to other species), we applied stringent filtering thresholds (see Materials and Methods) and identified both single nucleotide polymorphisms (SNPs) that varied between patients (Table S3) and intrahost single nucleotide variants (iSNVs) that varied within patients (Table S4). We found a total of 39 SNPs between patients and a range of 2 to 207 iSNVs within each patient metagenome (Table 1; Fig. 2). Given the wide variation in coverage across samples, we checked for any bias toward detecting iSNVs in high-coverage samples. We observed no correlation between the number of detected iSNVs and the depth of coverage (Fig. S3) ($\rho = -0.12$, $P = 0.65$, Pearson correlation), suggesting no coverage bias and that diversity levels are comparable across samples.

Several mechanisms could account for the origins of the observed iSNVs, including *de novo* mutation within a patient, coinfection by divergent *V. cholerae* strains, homologous recombination, or sequencing errors. iSNVs were distributed across the genome (Fig. 2A) rather than clustered in hot spots as would be expected if iSNVs arose from recombination events (28). Recombination thus appears to be an unlikely source of iSNVs, although further work is needed to confirm this. Despite stringent filters for iSNV calls in InStrain, some iSNVs could be false positives due to sequencing or read mapping errors. In patient E, sampled on two consecutive days, we detected 8 iSNVs on the first day, of which 4 were again detected on the second day, along with 13 additional iSNVs. It would be unlikely for random sequencing errors to occur in the exact same four sites on two consecutive days by chance alone, and therefore these iSNVs are likely either true positives or systematic (site-specific) sequencing or read mapping errors. However, systematic errors would be expected to be seen in other samples at the same nucleotide positions, which is not the case. The additional iSNVs detected at only one time point could be sequencing errors or could reflect iSNV allele frequency changes over time. In the analyses that follow, we acknowledge that a subset of iSNVs could be false positives but assume that this source of error is randomly distributed across samples and can thus be accounted for in statistical tests.

Although it is difficult to distinguish *de novo* mutation from coinfection using metagenomic data alone, the distribution of iSNV allele frequencies may provide clues. Specifically, under a standard neutral coalescent model, a single evolving population or

**TABLE 1** Within-patient *V. cholerae* diversity profiles from 15 metagenomes[a]

| Patient and/or day | Total no. of iSNVs | No. of nonsynonymous iSNVs | No. of synonymous iSNVs | No. of intergenic iSNVs | Mean coverage (×) | iRep value | DNA repair and proofreading genes with NS mutation |
|---|---|---|---|---|---|---|---|
| A | 93 | 6 | 0 | 87 | 451.3 | 3.34 | |
| B | 18 | 7 | 5 | 6 | 111.4 | 1.7 | |
| C | 6 | 0 | 1 | 5 | 111.8 | 1.7 | |
| D | 41 | 22 | 9 | 10 | 10 | 5.43 | DNA polymerase II |
| E | | | | | | | |
| Day 1 | 8 | 2 | 1 | 5 | 351 | 3.25 | |
| Day 2 | 21 | 7 | 1 | 13 | 258 | 1.23 | |
| F | 207 | 133 | 47 | 27 | 18.2 | 2.48 | DNA mismatch repair endonuclease MutL; nuclease SbcCD subunit C |
| G | 16 | 12 | 3 | 1 | 7.7 | 1.73 | |
| H | 32 | 21 | 11 | 0 | 98.5 | 4.75 | Excinuclease ABC subunit UvrB |
| I | 75 | 55 | 20 | 0 | 13 | 2.79 | MutT/nudix family protein |
| J | 6 | 1 | 0 | 5 | 424.6 | 1.84 | |
| K | 25 | 13 | 6 | 6 | 18 | 1.69 | Formamidopyrimidine-DNA glycosylase mutM |
| L | 13 | 9 | 1 | 3 | 164.4 | 2.67 | |
| M | 2 | 0 | 1 | 1 | 113 | 2.65 | |
| N | 7 | 2 | 1 | 3 | 6.7 | 2.27 | |

[a]Mutations segregating within patients are denoted iSNVs. The number of iSNVs and mean coverage values were computed with InStrain (27), and replication rates were determined with iRep (39).

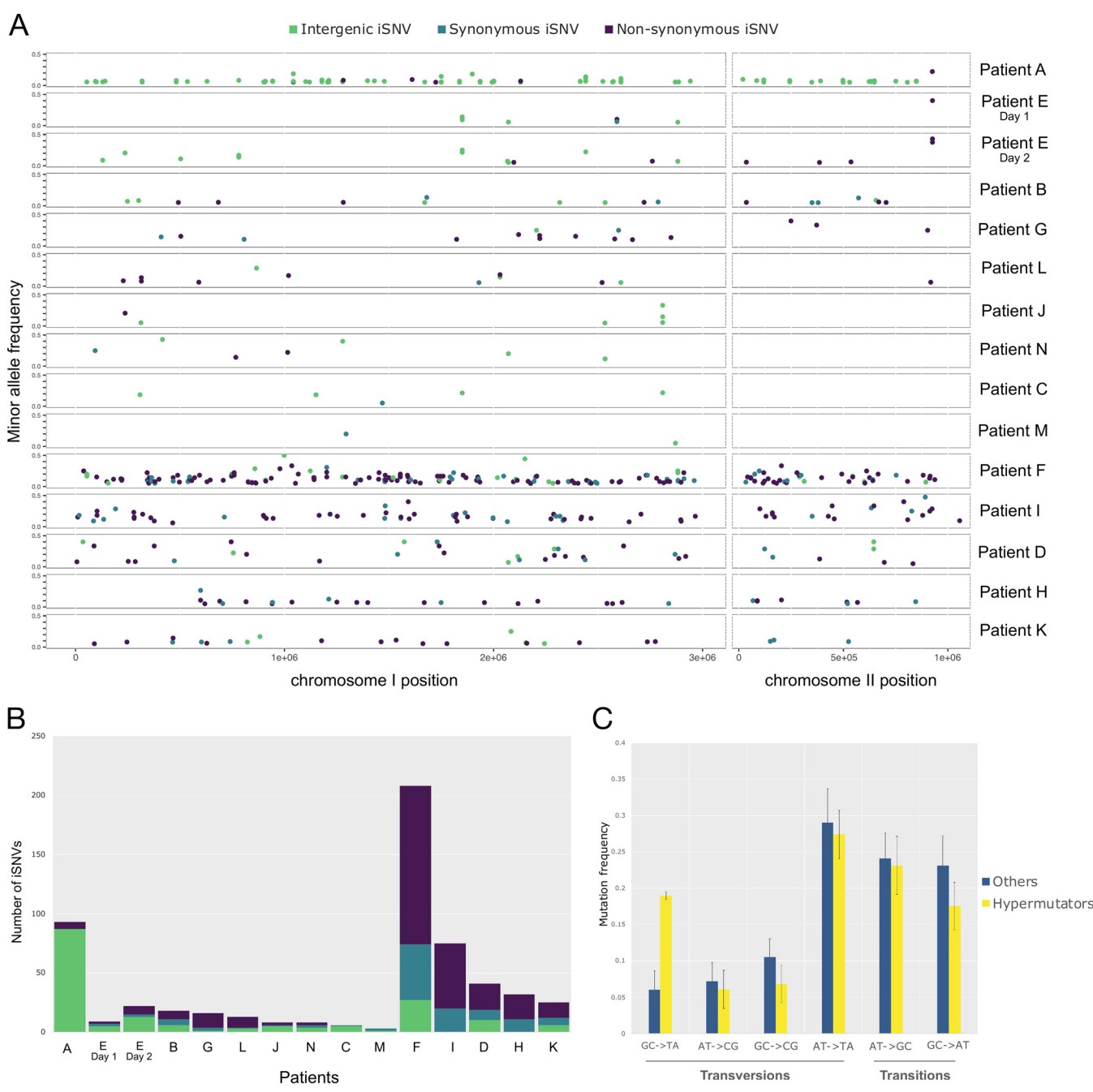

**FIG 2** Within-patient *Vibrio cholerae* diversity quantified from metagenomic data. (A) Minor allele frequency and distribution of intergenic, synonymous, and nonsynonymous iSNVs across the two *Vibrio cholerae* chromosomes for 14 patients with over 5× coverage of the *V. cholerae* genome. (B) Numbers of intergenic, synonymous, and nonsynonymous iSNVs for each patient. (C) Spectrum of within-patient mutation in hypermutators compared to nonmutators. Error bars represent standard errors of the mean. Only samples with 6 or more iSNVs were included to reduce noise from low counts.

strain is expected to generate a geometric distribution of iSNV frequencies, dominated by a peak of low-frequency mutations. In contrast, simple mixtures of a few strains (i.e., coinfections) will produce distributions with one or more peaks at intermediate allele frequencies (29). Examining the iSNV frequency distributions in our data revealed four patients (A, B, H, and L) dominated by a single peak of low-frequency alleles and two patients (F and I) with a possible peak at intermediate frequency, consistent with a mixture of strains at frequencies of roughly 0.15 and 0.85 (Fig. S1). Other patients had distributions too noisy to interpret, often due to small number of iSNVs. The iSNV frequency

distributions of patients F and I are consistent with coinfection but could equally be explained by within-host balancing or negative frequency-dependent selection. Coinfections might be expected to have lower ratios of nonsynonymous (NS) to synonymous (S) substitutions compared to within-host *de novo* mutations, since NS mutations are more likely to be deleterious and purged over evolutionary time. However, the NS:S ratios in patients F and I were in the same range as those observed in patients A, B, H, and L, providing no clear support for the coinfection hypothesis (Table 1). We therefore cannot exclude coinfection as a source of iSNVs in a minority of patients, although the evidence remains ambiguous.

**Evidence for *V. cholerae* hypermutators within patients.** In five of the six patients with a high number of iSNVs (>25), we identified nonsynonymous (NS) mutations in genes involved in DNA mismatch repair pathways, including the DNA polymerase II in patient D, or proteins of the methyl-directed mismatch repair (MMR) system in patients F, I, and K (Table 1). Assuming that DNA repair genes are of average length and contain an average number of NS sites, we can estimate the one-sided binomial probability that NS mutations occur in the observed number of DNA repair genes in each of these five patients (Table 1). We calculated this probability assuming a binomial success rate of 0.0127 (obtained by dividing 51, the number of DNA repair genes [GO:0006281] by 4,007, the total number of genes in the *V. cholerae* strain N16961 reference genome). By multiplying the probabilities from each patient, we obtain an overall probability of 0.0023 that we would see the observed number of DNA repair genes with NS mutations in all five patients. This number of patients with mutated DNA repair genes is therefore unlikely to have occurred by chance alone, given the observed number of mutations. We therefore defined these five patients as containing potential hypermutator lineages of *V. cholerae*.

Although the precise functional consequences of these NS mutations are unknown, they are potential loss-of-function mutations that could plausibly result in hypermutator phenotypes (17). In the patient harboring the highest number of variants (patient F, 207 iSNVs), we detected two NS mutations in two different genes coding for proteins involved in DNA repair: the DNA mismatch repair endonuclease MutL (17) and the nuclease SbcCD subunit C (24, 30, 31). Even with such a high number of iSNVs, it is surprising to observe NS mutations in two DNA repair genes in patient F (133 NS mutations in 4,007 genes; $P = 0.033$ for a mutation in one gene and $P = 0.0011$ for two genes). In patient I, in which we also detected a high number of iSNVs, an NS mutation in the gene coding for the MutT/nudix protein, involved in the repair of oxidative DNA damage (32), could also cause a strong hypermutation phenotype. Patients D, H, and K presented fewer iSNVs but also contained NS mutations in genes involved in DNA damage repair (33–35). However, some of these genes have been shown to play less critical roles in bacterial DNA repair than MutSLH (17, 36), which could lead to a weaker hypermutator phenotype.

The patient with the second highest number of iSNVs, patient A, contained a high number of intergenic variants (87 out of 96 iSNVs) (Fig. 2B) but no apparent NS mutations in genes involved in DNA repair; we therefore did not consider patient A a hypermutator. This large number of intergenic iSNVs could be caused by read mapping errors to a distantly related *V. cholerae* reference genome; however, the same iSNV calls were obtained when using the MAG from patient A as a reference genome. False iSNVs could also occur due to mismapping of reads from different species to the *V. cholerae* genome. Although we took measures to exclude such mismapping by removing reads mapped to 79 representative MAGs in our patient microbiomes, and by excluding sites with aberrant high or low depth of coverage (see Materials and Methods), we cannot exclude the possibility that patient A contained a cryptic member of the gut microbiome that resulted in mismapping.

Previous studies have noted mutational biases in hypermutators, such as an increase in transition over transversion mutations in a *Burkholderia dolosa* mutator with a defective MutL (18), or an excess of G:C→T:A transversions in a *Bacillus anthracis* hypermutator (37), and in members of the gut microbiome (38). When we compared the spectrum of

**TABLE 2** Set of genes with convergent mutations identified in more than one patient

| Protein (UniProt ID) | Mutation(s)$^a$ in: | | | | | | | |
|---|---|---|---|---|---|---|---|---|
| | Patient A | Patient B | Patient D | Patient E | Patient F | Patient H | Patient I | Patient K |
| Hemolysin (VC cytolysin) (P09545) | NS (0.22) | | | 3 NS (0.22-0.43) | | | | |
| 2-Aminoethylphosphonate ABC transporter ferric-binding protein (Q9KLY8) | | NS (0.05) | | NS (0.05) | | | | |
| Peptidase B (Q9KTX5) | | | NS (0.33) | | | | NS (0.09) | |
| Nuclease SbcCD subunit C (Q9KM67) | | | S (0.28) | | NS (0.09) | | | |
| C4-dicarboxylate transport sensor protein (Q9KN25) | | | | | NS (0.08) | | NS (0.11) | |
| Zinc/cadmium/mercury/lead-transporting ATPase (Q9KT72) | | | | | NS (0.08) | | | NS (0.06) |
| Hypothetical protein (A0A0H3AI44) | | | | | NS (0.14) | | | NS (0.14) |
| Hypothetical protein (Q9KLL1) | | | | | NS (0.33) | NS (0.11) | | |
| Formamidopyrimidine-DNA glycosylase mutM (C3LQI3) | | | | | S (0.18) | | | NS (0.08) |
| Phosphoribosylformylglycinamide synthase (Q9KTN2) | | | | | | | NS (0.06) | S (0.08) |

$^a$The presence of a synonymous or nonsynonymous iSNV in each gene and each patient is indicated by S or NS, respectively, and the minor allele frequency is shown in parentheses. None of the mutations were found at the same nucleotide or codon position. Underlined patient designations indicate patients containing likely hypermutators. Only genes and patients containing more than one mutated gene are shown.

mutations observed in suspected hypermutators to that of nonmutator samples, we found a significant difference (chi-square test, $P < 0.01$) due to an apparent excess of G:C→T:A transversions in hypermutators (Fig. 2C; Fig. S2). While not all NS mutations in DNA repair genes necessarily cause defects, we observed changes in the transition/transversion ratio concordant with the MMR gene mutated (Fig. S2). For instance, it has been shown in other bacterial pathogens that mutations in *mutT* and *mutL* lead to strong mutator phenotypes, increasing the rate of A:T→C:G transversions and G:C →A:T transitions, respectively (34), which we observed in patients (F and I) containing these mutations (Table 1; Fig. S2). Mutations in *mutM* were also previously associated with G:C →T:A mutations, as observed in patient K (Fig. S2). More experiments are clearly needed to confirm the phenotypes of these DNA repair mutants, but our results are largely consistent with known hypermutation profiles.

Current theory suggests that hypermutators may be adaptive under novel or stressful environmental conditions because they more rapidly explore the mutational space and are the first to acquire adaptive mutations. However, hypermutation comes at the cost of the accumulation of deleterious mutations. To test the hypothesis that hypermutation leads to fitness costs due to these deleterious mutations, we used iRep (39) to estimate *V. cholerae* replication rates in each sample and to test whether the replication rate was negatively associated with the number of iSNVs. iRep infers replication rates from MAGs and metagenomic reads (39). For instance, an iRep value of 2 would indicate that most of the population is replicating one copy of its genome. In our data (Table 1), iRep values varied from 1.23 (patient E at day 2) to 5.43 (patient D), and we did not find any association between the replication rate of *Vibrio cholerae* and the number of iSNVs detected within each subject (Fig. S3) (Pearson correlation, $\rho = 0.15$, $P > 0.05$). This lack of association could be due to noisy replication rate estimates from iRep and could be revisited in larger patient cohorts.

**Convergent evolution suggests adaptation of nonmutator *V. cholerae* within patients.** While none of the patients shared iSNVs at the exact same nucleotide position, some contained mutations in the same gene that occurred independently in more than one patient (Table 2). These are examples of convergent evolution at the gene level. To determine whether genes that acquired multiple mutations in independent patients could be under convergent selection within the host, we performed permutation tests for hypermutator and nonmutator samples separately (see Materials and Methods). This test identifies consistent signatures of either positive or relaxed purifying selection common to multiple hosts. Among the hypermutator samples, we identified five genes with NS mutations in two or more patients (Table 2), which was not an unexpectedly high level of convergence given the large number of mutations

in hypermutators (permutation test, $P = 0.97$). That the $P$ value approaches 1 suggests either that the hypermutators are actually selected against mutating the same genes in different patients or, more likely, that the permutation test is conservative. For the samples with no evidence of hypermutator phenotypes, we identified two genes with NS mutations in two patients. The first gene, *hlyA*, encodes a hemolysin that causes cytolysis by forming heptameric pores in human cell membranes (40), while the second gene encodes a putative ABC transporter ferric-binding protein (Table 2). Observing convergent mutations in two different genes is unexpected (permutation test, $P = 0.039$) in a test that is likely to be conservative. We also note that the three iSNVs in *hlyA* have relatively high minor allele frequencies (0.22 to 0.43) in comparison to other convergent NS mutations (median minor allele frequency of 0.11) (Table 2) and to NS mutations overall (median of 0.12) (Table S4). Together, these analyses suggest that *V. cholerae* hypermutators produce NS mutations that are predominantly deleterious or neutral. This does not exclude the possibility of adaptive mutations in hypermutators, but these are difficult to pinpoint against the overwhelming background of nonadaptive mutations. In contrast, nonmutators are subject to detectable within-patient positive selection on certain genes, which merits further investigation.

To further explore differential selection at the protein level within and between patients, we applied the McDonald-Kreitman test (41) to the 9 patients with no evidence for hypermutation and to the 5 patients harboring potential hypermutators. Based on whole-genome sequences of *V. cholerae* isolates, we previously found an excess of NS mutations fixed between patients in Bangladesh, based on a small sample of five patients (15). Here, based on metagenomes from a larger number of patients, we found the opposite pattern of a slight excess of NS mutations segregating as iSNVs within patients, consistent with slightly deleterious mutations occurring within patients and purged over evolutionary time. However, the difference between NS:S ratios within and between patients was not statistically significant (Fisher's exact test, $P > 0.05$) (Table S5); thus, the evidence for differential selective pressures within versus between cholera patients remains inconclusive.

Many NS mutations occurred in genes involved in transmembrane transport, pathogenesis, response to antibiotics, secretion systems, chemotaxis, and metabolic processes (Fig. S4). Both hypermutator samples (Fig. S4B) and nonmutators (Fig. S4C) have a high NS:S ratio in genes of unknown function, while hypermutators have many NS mutations in transmembrane proteins, which are absent in nonmutators. However, nonmutator samples have more NS mutations in genes involved in pathogenesis and secretion systems. Most of the NS mutations involved in pathogenesis were found in the gene *hlyA* (a target of convergent evolution, mentioned above).

**Evidence for a hypermutator from isolate whole-genome sequencing.** In addition to metagenomic analyses, we performed whole-genome sequencing of multiple *Vibrio cholerae* clinical isolates from index cases and asymptomatic contacts (Fig. 1A) from three households (56, 57, and 58) (Table S1). As noted above, asymptomatic infected contacts did not yield sufficient metagenomic reads to assemble the *V. cholerae* genome or call iSNVs, but their stool cultures yielded colonies for whole-genome sequencing. The first asymptomatic contact, 58.01, tested positive for *Vibrio cholerae* on day 4 after the presentation of the index case to the hospital, and *Vibrio cholerae* was cultured from the stool on days 4, 6, 7, and 8. We sequenced five isolates respectively from day 4 and 6 samples and four isolates from each of the subsequent days. For households 56 and 57, five isolates were sequenced from each sample, at day 1 for the index cases and at day 2 for the asymptomatic carriers (Table S6).

The index case from household 58 (patient N) was the only sample also included in the metagenomic analysis described above, allowing a comparison between culture-dependent and -independent assessments of within-patient diversity. We did not detect any iSNVs among the five isolates sequenced from patient N. In contrast, the metagenomic analysis of patient N revealed seven iSNVs (Table 1), suggesting a higher sensitivity for the detection of rare variants which could be easily missed by

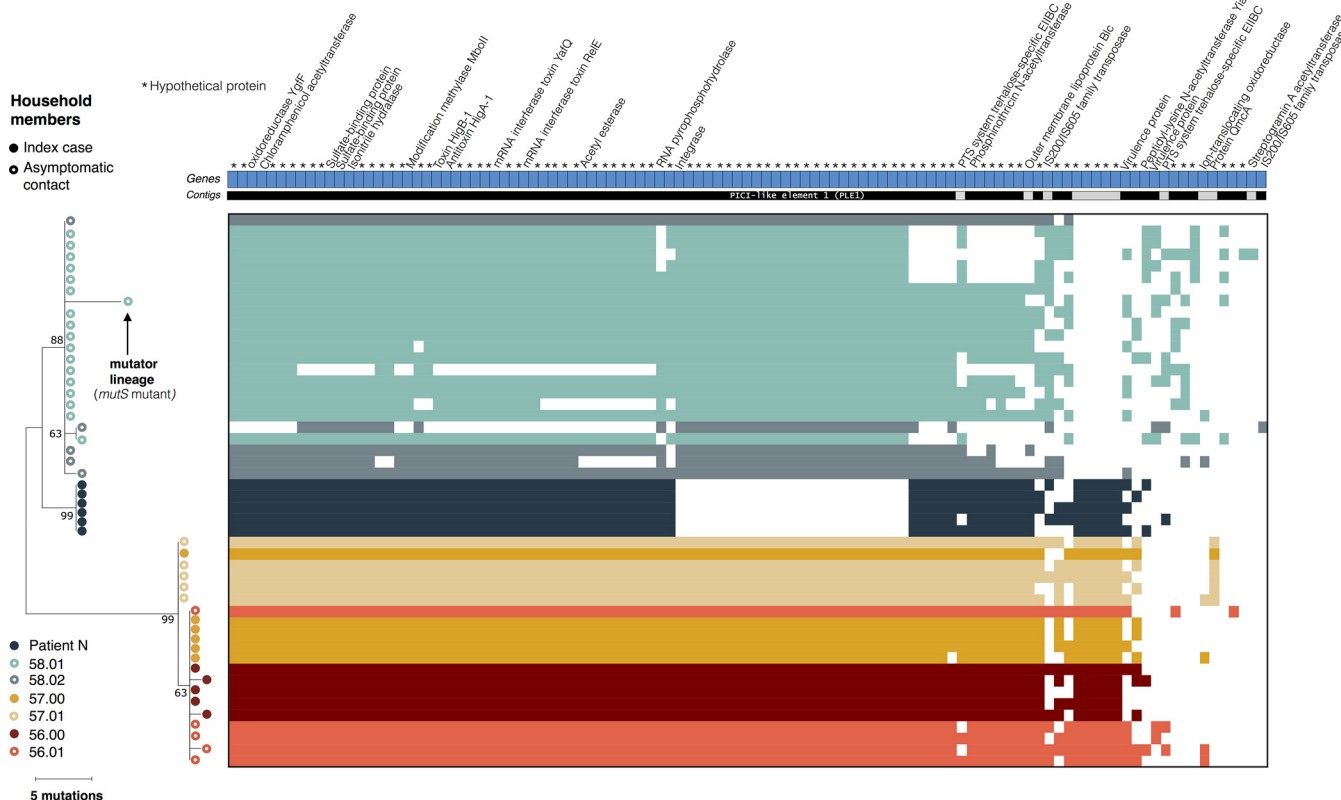

**FIG 3** Phylogeny and pangenome of 48 *Vibrio cholerae* isolates from index cases and their asymptomatic contacts. The phylogeny was inferred using maximum parsimony. The percentages of replicate trees in which the associated taxa clustered together in the bootstrap test (1,000 replicates) are shown next to the branches. Filled circles represent isolates from index cases, and empty circles represent isolates from their asymptomatic contacts. Patient N is the index case of household 58. All other index cases are labeled ".00", with household contacts labeled ".01" and ".02". The heat map of gene presence-absence is based on 106 genes in the flexible genome. Colored blocks in the heat map indicate gene presence; white indicates gene absence. Each row corresponds to an isolate from the phylogenetic tree, and each column represents an orthologous gene family. Each unique color represents a different individual. Different contigs are represented as alternating blocks of black and light gray.

sequencing only a few isolates. Despite a potentially higher error rate, metagenomics is more appropriate for sensitively detecting iSNVs when only shallow isolate sequencing is possible. Distinguishing between these possibilities would require sequencing a much larger number of colonies, which is beyond the scope of the present study.

In contrast to metagenomes consisting of many unlinked reads, whole-genome sequencing allows the reconstruction of a phylogeny describing the evolution of *V. cholerae* within and between patients based on SNVs in the core genome (Fig. 3). As described previously (6), isolates from members of the same household tended to cluster together. In index case 57.00, all isolates were identical in terms of SNVs, with the exception of one isolate that was identical to the five isolates sequenced from the asymptomatic contact from the same household, patient 57.01 (Table 3; Fig. 3). In the inferred phylogeny, isolates from contact 57.01 are ancestral, and the more common genotype in index case 57.00 has one additional derived mutation. This shared ancestral genotype between the two individuals was unexpected and might suggest a potential transmission event from the asymptomatic contact to the index case, followed by a mutational event and the spreading of the new variant in the index case. The only mutation found in four of the five isolates from the index case was a nonsynonymous mutation in a gene coding for a cyclic-di-GMP-modulating response regulator, which could have an impact on the regulation of biofilm formation in the host (42). However, the direction of transmission (from contact to index case) is supported only by one mutation and therefore remains uncertain. The genomes from household 57 are also similar or identical to genomes from household 56, suggesting further caution in inferring transmission chains. Among the other index cases with isolate genome sequences, we found no iSNVs in

**TABLE 3** Nucleotide changes identified in core genes of the *V. cholerae* isolates from index cases [patients 56.00, 57.00, and 58.00 (N)] and their asymptomatic contacts[a]

| Type | Isolate(s) | Nucleotide position in MJ-1236 | Mutation type | Ref nucleotide | Alt nucleotide | Gene annotation | Patients with metagenomic samples with same variant |
|---|---|---|---|---|---|---|---|
| iSNV | 58.01d7C1 | Chr1, 53054 | NS | G | A | DNA mismatch repair protein MutS | |
| SNP | Households 56 and 57 | Chr1, 198988 | S | G | A | MSHA biogenesis protein MshQ | |
| iSNV | 58.01d7C1 | Chr1, 209665 | NS | G | A | MSHA biogenesis protein MshN | |
| iSNV | 56.00C4 | Chr1, 374172 | NS | C | T | UDP-*N*-acetylglucosamine 4,6-dehydratase | |
| SNP | Household 58 | Chr1, 410638 | NS | G | A | Phosphopantetheine adenylyltransferase | M, N |
| SNP | Households 56 and 57 | Chr1, 754154 | NS | C | T | 1,4-Dihydroxy-2-naphthoate polyprenyltransferase | |
| SNP | Household 58 | Chr1, 841538 | S | C | T | SSU ribosomal protein S4p | L, M, N |
| SNP | Household 58 | Chr1, 1315021 | S | T | G | Exported zinc metalloprotease YfgC precursor | L, M, N |
| iSNV | 58.02C1 | Chr1, 1576083 | S | C | A | Periplasmic thiol:disulfide oxidoreductase DsbB | |
| SNP | Patient N | Chr1, 1689779 | NS | A | C | Sigma-54 dependent transcriptional regulator | |
| SNP | Contacts 58.01 and 58.02 | Chr1, 2301641 | NS | G | A | Putative membrane protein | |
| iSNV | 58.01d7C1 | Chr1, 1744854 | NS | C | T | Hypothetical protein | |
| SNP | Contacts 58.01 and 58.02 | Chr1, 2262202 | NS | A | G | Serine transporter | |
| SNP | Households 56 and 57 | Chr1, 2301641 | NS | C | T | LacI family DNA-binding transcriptional regulator | D, J, K |
| iSNV | 57.00C5 | Chr1, 2509468 | NS | C | T | Cyclic-di-GMP-modulating response regulator | |
| iSNV | 56.01C1 | Chr1, 2588496 | NS | C | T | Amidophosphoribosyltransferase | |
| iSNV | 58.01d7C1 | Chr1, 2693815 | NS | C | T | PTS system, trehalose-specific IIB component | |
| SNP | Household 58 | Chr1, 2806858 | NS | A | T | Citrate lyase alpha chain | L, M, N |
| iSNV | 56.00C1 | Chr1, 3037471 | S | A | G | Hypothetical protein | |
| SNP | Patient N | Chr1, 3059131 | NS | C | T | DNA polymerase V (UmuC) | |
| SNP | Households 56 and 57 | Chr1, 3095039 | NS | G | A | Outer membrane protein OmpU | D, F, G, I, K |
| SNP | Contacts 58.01 and 58.02 | Chr1, 3105102 | S | C | T | Glutamate-1-semialdehyde aminotransferase | |
| iSNV | 58.01d7C1 | Chr2, 528409 | NS | C | T | Vibriolysin, extracellular zinc protease | |

[a]Genome position is according to the MJ-1236 reference genome (CP001485.1, CP001486.1). Mutations segregating within patients are denoted iSNVs; mutations fixed between patients are denoted SNPs. SNPs fixed within all members of one or more households are also designated household SNPs. Patient allele frequency shows the allele frequency of the alternative (minor) allele. Ref, reference allele; Alt, alternative allele; NS, nonsynonymous; S, synonymous; Chr1, chromosome 1; Chr2, chromosome 2; MSHA, mannose-sensitive hemagglutinin; SSU, small subunit; PTS, phosphotransferase system.

**TABLE 4** Flexible gene content variation within and between patients[a]

| Patient | No. of genes fixed within patients | No. of genes variable within patients | No. of singletons |
|---|---|---|---|
| 56.00 | 88 | 6 | 0 |
| 56.01 | 86 | 10 | 0 |
| 57.00 | 87 | 8 | 0 |
| 57.01 | 87 | 8 | 0 |
| 58.00 (N) | 62 | 9 | 0 |
| 58.01 | 36 | 65 | 2 |
| 58.02 | 39 | 67 | 1 |

[a]Singletons are defined as genes found in only one isolate and are also counted as variable genes within patients. Genes fixed within patients are present in all isolates from a patient but are absent in at least one other isolate in the study.

patient N and two iSNVs in patient 56.00. One isolate from this patient had a synonymous mutation in a hypothetical protein, and another isolate had a nonsynonymous mutation in a UDP-*N*-acetylglucosamine 4,6-dehydratase gene (Table 3). We detected iSNVs in the other asymptomatic contacts, with one synonymous and one intergenic mutation in contact 58.02 and one nonsynonymous mutation in one isolate from contact 56.01 (Table 3; Fig. 3).

Notably, we also found evidence for a hypermutator in contact 58.01. One isolate sampled from this contact had the highest number of mutations seen in any branch in the phylogeny (five NS mutations, all G:C→T:A transversions) relative to its ancestral branch (i.e., to the other isolates from the same person). This high mutation rate could be explained by an NS mutation in the gene encoding MutS, another key component of the methyl-directed mismatch repair (MMR) system (Table 3; Fig. 3). The mutation in this gene could explain the accumulation of a surprising number of mutations in this isolate, which is likely a hypermutator with a characteristic transversion bias. This contact contained no SNVs among the isolates sampled on days 4 and 6, and we found this likely hypermutator isolate on day 7. However, the hypermutator was not observed again at day 8, due either to the lower resolution in the detection of variants with the WGS of cultured isolates or the disappearance of this mutant from the population.

**Pangenome analyses.** Whole-genome isolate sequencing also provides the opportunity to study variation in gene content (the pangenome) within and between patients. We identified a total of 3,478 core genes common to all *V. cholerae* genomes and 106 flexible genes present in some but not all genomes (Fig. 3; Table 4). We also found an additional 251 genes present uniquely in isolate 56.00C4, assembled into one single contig identified as the genome of the lytic *Vibrio* phage ICP1, which was assembled alongside the *Vibrio cholerae* genome. This phage contig contained the ICP1 CRISPR/Cas system, which consists of two CRISPR loci (designated CR1 and CR2) and six *cas* genes, as previously described (43, 44). These genes were excluded from subsequent *V. cholerae* pangenome analyses.

Among the 106 flexible genes, some varied in the presence/absence within a patient, ranging from 36 to 88 genes gained or lost per patient (Table 4; Fig. 3). The majority of these flexible genes (75%) were annotated as hypothetical, and several were transposase or prophage genes. A large deletion of 24 genes was detected in the isolates from patient N, in an 18-kb phage-inducible chromosomal island (PICI) previously shown to prevent phage reproduction and which is targeted by the ICP1 CRISPR/Cas system (44). These PICI-like elements are induced during phage infection and interfere with phage reproduction via multiple mechanisms (45, 46). The deletion of this PICI element in the *V. cholerae* genome may be a consequence of an ongoing evolutionary arms race between *V. cholerae* and its phages.

## DISCUSSION

Although within-patient *Vibrio cholerae* genetic diversity has been reported previously (14, 15, 47, 48), our results confirmed that within-patient diversity is a common

feature observed in symptomatic patients with cholera but also in a small sample of asymptomatically infected individuals. In this study, we used a combination of metagenomic sequencing and WGS technologies to characterize this within-patient diversity, revealing evidence for hypermutator phenotypes in both symptomatic and asymptomatic infections.

In our previous study, we detected between zero and three iSNVs in cultured isolates from patients with acute infection (15). In contrast, metagenomic analyses allowed us to detect 2 iSNVs in the patient with the lowest level of diversity but up to 207 iSNVs in another individual (Table 1). In the only patient for which we were able to characterize *Vibrio cholerae* intrahost diversity both from the metagenome and from cultured isolates, we did not identify any iSNVs among five sequenced isolates but detected 7 iSNVs from the metagenomic analyses. This could be due to false-positive iSNV calls inferred from metagenomes but might also represent a higher sensitivity to identify rare mutations (49). Our previous phylogenetic analysis of *V. cholerae* isolate genomes also concluded that within-patient mutation was a more likely source of variation than coinfection with multiple strains of *V. cholerae* (15). Although neither isolate WGS nor metagenomic analysis can fully exclude the possibility of coinfection (especially involving very closely related strains), neither our earlier nor our present study provides strong evidence for coinfection.

Despite its potential sensitivity to detect rare variants (26), metagenomics has limitations. As already mentioned, some of the iSNVs inferred from metagenomes could be false positives, and this deserves further benchmarking. Within-sample diversity profiles cannot be established for low-abundance microbes with insufficient sequence coverage ($<5\times$) and depth, and this level of coverage is difficult to obtain in diverse microbial communities. In this study, only 48% of the samples from patients with acute symptoms, known to harbor a high fraction of vibrios in their stool ($10^{10}$ to $10^{12}$ vibrios per liter of stool), contained enough reads to reconstruct *Vibrio cholerae* MAGs and to quantify within-patient diversity. Asymptomatic patients typically shed even less *V. cholerae* in their stool (50), making it even more challenging to assemble their genomes using metagenomics without depletion of host DNA or targeted sequence capture techniques (51, 52).

Hypermutation is defined as an excess of mutations due to deficiency in DNA mismatch repair, and hypermutator strains have been described in diverse pathogenic infections and *in vivo* experiments, including *Pseudomonas aeruginosa*, *Haemophilus influenzae*, and *Streptococcus pneumoniae* in cystic fibrosis patients or *Escherichia coli* in diverse habitats (17, 34, 53). In *Vibrio cholerae*, a previous study of 260 clinical isolate genomes identified 17 isolates with an unusually high number of SNPs uniformly distributed along the genome (24). Most of these genomes contained mutations in one or more of four genes (*mutS*, *mutH*, *mutL*, and *uvrD*) that play key roles in DNA mismatch repair (24). The authors of that study cautiously suggested that this apparent high frequency of hypermutators could be associated with the rapid spread of the seventh cholera pandemic, particularly because hypermutators may be a sign of population bottlenecks and recent selective pressure. However, they also hypothesized that these high mutation rates could be artefactual because the *V. cholerae* isolates had been maintained in stab cultures for many years. It thus remains unclear if a hypermutator phenotype was derived within patients or during culture (24, 25). Using our metagenomic approach, we provide evidence that hypermutators can indeed emerge during infection, because DNA was extracted directly from patient samples without a culture step in which mutations could have occurred during DNA replication. Using culture-based WGS, with only a brief overnight culture, we report further evidence that hypermutators occur in asymptomatic patients as well. An alternative explanation is that hypermutators evolved very recently prior to infection, a possibility which is difficult to exclude or test. Even if the number of iSNVs is somewhat inflated by metagenomic sequencing, the concordance of known DNA repair mutations with a transversion-skewed mutation profile is consistent with current knowledge of hypermutation

and not easily explained by sequencing errors alone. Future work is required to determine any impacts of hypermutation on cholera disease severity or transmission.

Hypermutator phenotypes are believed to be advantageous for the colonization of new environments or hosts, allowing the hypermutator bacteria to generate adaptive mutations more quickly, which leads to the more efficient exploitation of resources or increased resistance to environmentally stressful conditions, such as antibiotics (17, 20, 34, 53). However, this high mutation rate can have a negative impact on fitness in the long term, with most of the mutations being neutral or deleterious (20, 22, 54). A mouse model study showed that hypermutation can be an adaptive strategy for *V. cholerae* to resist host-produced reactive oxygen-induced stress and lead to a colonization advantage by increased catalase production and increased biofilm formation (23). In our study of convergent evolution, we found no evidence for adaptive mutations in the hypermutators. This could be because the signal from a small number of adaptive mutations is obscured by overwhelming noise from a large number of neutral or deleterious mutations. Further work is therefore needed to determine if *V. cholerae* mutators produce adaptive mutations during human infection.

In contrast, we did find evidence for an excess of convergent mutations occurring independently in the same genes in different patients, suggesting parallel adaptation in nonmutator *V. cholerae* infections. Specifically, two patients contained mutations in the same hemolysin gene, *hlyA*, which codes for a toxin that has both vacuolating and cytocidal activities against a number of cell lines, including human intestinal cells (55), and is known to be an important virulence factor in *Vibrio cholerae* El Tor O1 and a major target of immune responses during acute infection (56, 57). Previous studies of within-patient *V. cholerae* evolution did not identify mutations in *hlyA* and instead identified different mutations possibly under selection for biofilm formation (15) or phage resistance phenotypes (14). This lack of concordance might be explained by relatively modest sample sizes of cholera patients in these studies but could also suggest that selective pressures may be idiosyncratic and person specific across *Vibrio cholerae* infections.

In conclusion, our results illustrate the potential and limitations of metagenomics as a culture-independent approach for the characterization of within-host pathogen diversity. We also provide evidence that hypermutators emerge within human *V. cholerae* infection, and their evolutionary dynamics and relevance to disease progression merit further study.

## MATERIALS AND METHODS

**Ethical statement.** The Ethical and Research Review Committees of the icddr,b (International Center for Diarrheal Disease Research, Bangladesh) and the Institutional Review Board of MGH reviewed the study. All adult subjects provided informed consent, and parents/guardians of children provided informed consent. Informed consent was written.

**Sample collection, clinical outcomes, and metagenomic sequencing.** To study within-host diversity of *V. cholerae* during infection, we used stool and rectal swab samples collected from cholera patients admitted to the icddr,b Dhaka Hospital, and from their household contacts, as previously described (12). Patients present to the icddr,b year-round with cholera, and cases peak during biannual floods (58, 59). Index cases were defined as patients presenting to the hospital with severe acute diarrhea and a stool culture positive for *V. cholerae*. Individuals who shared the same cooking pot with an index patient for 3 or more days are considered household contacts and were enrolled within 6 h of the presentation of the index patient to the hospital. Rectal swabs were collected each day during a 10-day follow-up period after presentation of the index case. Household contacts underwent daily clinical assessment of symptoms and collection of blood for serological testing. Contacts were determined to be infected if any rectal swab culture was positive for *V. cholerae* or if the contact developed diarrhea and a 4-fold increase in vibriocidal titer during the follow-up period (10, 11). If they developed watery diarrhea during the follow-up period, contacts with positive rectal swabs were categorized as symptomatic and those without diarrhea were considered asymptomatic. We excluded patients of ages below 2 and above 60 years old or with major comorbid conditions (10, 11).

Fecal samples and rectal swabs from the day of infection and follow-up time points were collected and immediately placed on ice after collection and stored at −80°C until DNA extraction. DNA extraction was performed with PowerSoil DNA extraction kits (Qiagen) after preheating to 65°C for 10 min and to 95°C for 10 min. Sequencing libraries were constructed for 33 samples from 31 patients, for which we obtained enough DNA. We used the NEBNext Ultra II DNA library prep kit and sequenced the libraries

on the Illumina HiSeq 2500 (paired-end 125 bp) and the Illumina NovaSeq 6000 S4 (paired-end 150 bp) platforms at the Genome Québec sequencing platform (McGill University).

**Metagenomic analyses. (i) Sequence preprocessing and assembly.** Sequencing fastq files were quality checked with FastQC (https://www.bioinformatics.babraham.ac.uk/projects/fastqc/). We removed human and technical contaminant DNA by aligning reads to the PhiX genome and the human genome (hg19) with Bowtie2 (60), and used the iu-filter-quality-minoche script of the illumina-utils program with default parameters to filter the reads (61).

**(ii) Taxonomic assignment.** Processed paired-end metagenomic sequences were classified using two taxonomic profilers: Kraken2 v.2.0.8_beta (a k-mer matching algorithm) (62) and MIDAS v.1.3.0 (a read mapping algorithm) (63). Kraken 2 examines the k-mers within a query sequence, uses the information within those k-mers to query a database, and then maps k-mers to the lowest common ancestor (LCA) of all genomes known to contain a given k-mer. Kraken2 was run against a reference database containing all RefSeq viral, bacterial, and archaeal genomes (built in May 2019), with default parameters. MIDAS uses a panel of 15 single-copy marker genes present in all ~31,000 bacterial species included in its database to perform taxonomic classification and maps metagenomic reads to this database to estimate the read depth and relative abundance of 5,952 bacterial species. We identified metagenomic samples containing *V. cholerae* and vibriophage reads and computed the mean depth of coverage (number of reads per base pair) of the *V. cholerae* pangenome in the MIDAS database (Table 1).

**(iii) Assembly and binning of *Vibrio cholerae* genomes.** To recover good-quality metagenome-assembled genomes (MAGs) of *V. cholerae*, we selected metagenomic samples with coverage of >10× against the *V. cholerae* pangenome in the MIDAS database and used MEGAHIT v.1.2.9 (64) to perform *de novo* assembly. For 9 of the 11 selected samples, we independently assembled the genome of each sample and coassembled the two remaining samples, which belonged to the same patient (a symptomatic infected contact on days 9 and 10). Contigs of <1.5 kb were discarded.

We extracted MAGs by binning our metagenomic assemblies. Because no single binning approach is superior in every case, with performance of the algorithms varying across samples, we used different binning tools to recover MAGs. The quality of a metagenomic bin is evaluated by its completeness (the level of coverage of a population genome) and the contamination level (the amount of sequence that does not belong to this population from another genome). These metrics can be estimated by counting the frequency of single-copy marker genes within each bin (65). We inferred bins using CONCOCT v.1.1.0 (66), MaxBin 2 v.2.2.7 (67), and MetaBAT 2 v.2.12.1 (68), with default parameters. We then used DAS_Tool v.1.1.1 on the results of these three methods to select a single set of nonredundant, high-quality bins per sample (69). DAS_Tool is a bin consolidation tool which predicts single-copy genes in all the provided bin sets, aggregates bins from the different binning predictions, and extracts a more complete consensus bin from each aggregate such that the resulting bin has the most single-copy genes while having a reasonably low number of duplicate genes (69). We then used Anvi'o v.6.1 (70) to manually refine the bins with contamination higher than 10% and Centrifuge v.1.0.4_beta (71) to determine the taxonomy of all bins in each sample, in order to identify *V. cholerae* MAGs.

Bins with completeness of >60% and contamination of <10% were first selected, and those assigned to *V. cholerae* were further filtered (completeness of >90% and contamination of <1% for the *V. cholerae* bins). We dereplicated the entire set of bins with dRep v.2.2.3 using a minimum completeness of 60%, the ANImf algorithm, 99% secondary clustering threshold, a maximum contamination of 10%, and a 25% minimum coverage overlap and obtained 79 MAGs displaying the best quality and representing individual metagenomic species (MGS).

**(iv) Detection of *Vibrio cholerae* genetic diversity within and between metagenomic samples.** We created a Bowtie2 index of the 79 representative genomes from the dereplicated set, including a single high-quality *Vibrio cholerae* MAG, and mapped reads from each sample to this set. By including many diverse microbial genomes in the Bowtie2 index, we aimed to avoid the mismapping of reads from other species to the *V. cholerae* genome and to reduce potential false-positive intrahost single nucleotide variant (iSNV) calls. As recommended, we used *Vibrio cholerae* MAGs from the samples under study rather than a genetically distant reference, as read mapping to the most closely related genome available is expected to reduce the rate of false-positive iSNV calls (72). We mapped the metagenomics reads of each sample with a *V. cholerae* coverage value of >5× (obtained with MIDAS) against the set of 79 MAGs, using Bowtie2 (60) with the –very-sensitive parameters. We also used Prodigal (73) on the concatenated MAGs, in order to predict open reading frames using default metagenomic settings.

We then used InStrain on the 15 selected samples (https://instrain.readthedocs.io/en/latest/index.html). This program aims to identify and compare the genetic heterogeneities of microbial populations within and between metagenomic samples (27). "InStrain profile" was run on the mapping results, with the minimum percent identity of read pairs to consensus set to 99%, the minimum depth of coverage to call a variant of 5×, and the minimum allele frequency to confirm a SNV equal to 0.05. All nonpaired reads were filtered out, as well as reads with an identity value below 0.99. Coverage and breadth of coverage (percentage of reference base pairs covered by at least one read) were computed for each genome. InStrain identified both biallelic and multiallelic SNV frequencies at positions where phred30 quality-filtered reads differ from the reference genome and at positions where multiple bases were simultaneously detected at levels above the expected sequencing error rate. SNVs were classified as nonsynonymous, synonymous, or intergenic based on gene annotations, and gene functions were recovered from the UniProt database (74) and BLAST (75). Then, filters similar to those described in reference 29 were applied to the detected SNVs. We excluded from the analysis positions with a very low or high coverage value, $C$, compared to the median coverage, $\overline{C}$, and positions within 100 bp of contig extremities. As sites with very low coverage could result from a bias in sequencing or library preparation

and sites with higher coverage could arise from mapping errors or be the result of repetitive region or multicopy genes not well assembled, we masked sites in all the samples if $C$ was $<0.3\bar{C}$ and if $C$ was $>3\bar{C}$ in at least two samples.

**(v) Mutation spectrum of hypermutator and nonmutator samples.** For each sample, iSNVs were categorized into six mutation types based on the chemical nature of the nucleotide changes (transitions or transversions). We combined all the samples with hypermutators and compared them to the mutation spectrum of the nonmutators. The mutation spectrum was significantly different between the hypermutator samples and the nonhypermutator samples (chi-square test, $P < 0.01$). We then computed the mutation mean and standard error of each of the six mutation types and compared the two groups (Fig. 2C).

**(vi) Bacterial replication rate estimation.** Replication rates were estimated with the metric iRep (index of replication), which is based on the measurement of the rate of decrease in average sequence coverage from the origin to the terminus of replication. iRep values (39) were calculated by mapping the sequencing reads of each sample to the *V. cholerae* MAG assembled from that sample.

**(vii) Tests for natural selection.** First, we identified signals of convergent evolution in the form of nonsynonymous iSNVs occurring independently in the same gene in multiple patients. To assess the significance of convergent mutations, we compared their observed frequencies to expected frequencies in a simple permutation model. We ran separate permutations for nonmutators (two genes with convergent mutations in at least two out of eight nonmutator samples, including only one time point from the patient sampled twice and excluding the outlier patient A with a large number of intergenic iSNVs) and possible hypermutators (five genes with convergent mutations in at least two out of five possible hypermutator samples). In each permutation, we randomized the locations of the nonsynonymous mutations, preserving the observed number of nonsynonymous mutations in each sample and the observed distribution of gene lengths. For simplicity, we assumed that two of three nucleotide sites in coding regions were nonsynonymous. We repeated the permutations 1,000 times and estimated a $P$ value as the fraction of permutations yielding greater than or equal to the observed number of genes mutated in two or more samples.

Second, we compared natural selection at the protein level within versus between patients, using the McDonald-Kreitman test (41). We again considered hypermutators separately. Briefly, the four counts ($P_n$, $P_s$, $D_n$, $D_s$) of between-patient divergence ($D$) versus within-patient polymorphism ($P$), and nonsynonymous (n) versus synonymous (s) mutations were computed and tested for neutrality using a Fisher exact test (false discovery rate [FDR] corrected $P$ values of $<0.05$).

**Whole-genome sequencing analyses. (i) Culture of *Vibrio cholerae* isolates.** We selected three of the households with asymptomatic infected contacts (households 56, 57, and 58) for within-patient diversity analysis using multiple *V. cholerae* colonies per individual. Each index case was sampled on the day of presentation to the icddr,b, and asymptomatic contacts positive for *V. cholerae* were sampled on the following day, except for one contact (household 58, contact 02). This individual was positive only on day 4 following presentation of the index case, and we collected samples and cultured isolates from day 4 to day 8. Stool samples collected from three index cases and their respective infected contacts were streaked onto thiosulfate-citrate-bile salts-sucrose (TCBS) agar, a medium selective for *V. cholerae*. After overnight incubation, individual colonies were inoculated into 5 ml Luria-Bertani broth and grown at 37°C overnight. For each colony, 1 ml of broth culture was stored at −80°C with 30% glycerol until DNA extraction. We used the Qiagen DNeasy blood and tissue kit, using 1.5 ml bacteria grown in LB medium, to extract the genomic DNA. In order to obtain pure genomic DNA (gDNA) templates, we performed an RNase treatment, followed by purification with the MoBio PowerClean pro DNA cleanup kit.

**(ii) Whole-genome sequencing and preprocessing.** We prepared 48 sequencing libraries using the NEBNext Ultra II DNA library prep kit (New England Biolabs) and sequenced them on the Illumina HiSeq 2500 (paired-end 125 bp) platform at the Genome Québec sequencing platform (McGill University). Sequencing fastq files were quality checked with FastQC, and Kraken2 was used to test for potential contamination with other bacterial species (62).

**(iii) Variant calling and phylogeny.** We mapped the reads for each sample to the MJ-1236 reference genome and called single nucleotide polymorphisms (SNPs; fixed within patients) and single nucleotide variants (SNVs; variable within patients) using Snippy v.4.6.0 (76), with default parameters. A concatenated alignment of these core variants was generated, and an unrooted phylogenic tree was inferred using maximum parsimony (MP) in MEGA X (77). The percentages of replicate trees in which the associated taxa clustered together in the bootstrap test (1,000 replicates) are shown next to the branches. The MP tree was obtained using the subtree-pruning-regrafting (SPR) algorithm with search level 1, in which the initial trees were obtained by the random addition of sequences (10 replicates).

**(iv) *De novo* assembly and pangenome analyses.** We *de novo* assembled genomes from each isolate using SPAdes v.3.14 on the short reads, with default parameters (78), and used Prokka v1.14.6 (79) to annotate them. We constructed a pangenome from the resulting annotated assemblies by combining Roary v.3.13.0 (80) and GenAPI (81), identifying genes present in all isolates (core genome) and genes present only in some isolates (flexible genome). The flexible genome and the phylogenetic tree were visualized with Phandango v.1.1.0 (82).

**Data availability.** All metagenomic sequence data are available in NCBI GenBank under BioProject PRJNA668607, and isolate genome sequences are available under BioProject PRJNA668606.

## SUPPLEMENTAL MATERIAL

Supplemental material is available online only.

**FIG S1**, TIF file, 1.7 MB.

**FIG S2**, TIF file, 0.7 MB.

**FIG S3**, TIF file, 0.3 MB.
**FIG S4**, TIF file, 1.4 MB.
**TABLE S1**, XLSX file, 0.01 MB.
**TABLE S2**, XLSX file, 0.01 MB.
**TABLE S3**, XLSX file, 0.01 MB.
**TABLE S4**, XLSX file, 0.1 MB.
**TABLE S5**, XLSX file, 0.01 MB.
**TABLE S6**, XLSX file, 0.01 MB.

## ACKNOWLEDGMENTS

We are grateful to the people of Dhaka, where our study was undertaken, to the field, laboratory, and data management staff, who provided a tremendous effort to make the study successful, and to the people who provided valuable support in our study. The icddr,b gratefully acknowledges the Government of the People's Republic of Bangladesh, Global Affairs Canada (GAC), the Swedish International Development Cooperation Agency (SIDA), and the Department for International Development (UKAid).

We declare that there are no conflicts of interest.

This study was supported by a Canadian Institutes of Health Research Operating Grant to B.J.S.; by the icddr,b Centre for Health and Population Research; by grants AI103055 (J.B.H. and F.Q.), AI106878 (E.T.R. and F.Q.), AI058935 (E.T.R., S.B.C., and F.Q.), and T32A1070611976 and K08AI123494 (A.A.W.) and by an Emerging Global Fellowship Award TW010362 (T.R.B.) from the National Institutes of Health; and by the Robert Wood Johnson Foundation Harold Amos Medical Faculty Development Program (R.C.C.).

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
