## [Reviewer comments · mSystems]

A combination of metagenomic and cultivation approaches reveals hypermutator phenotypes within *Vibrio cholerae* infected patients

Ines Levade, Ashraful Khan, Fahima Chowdhury, Stephen B. Calderwood, Edward Ryan, Jason Harris, Regina LaRocque, Taufiqur Bhuiyan, Firdausi Qadri, Ana Weil, and B. Jesse Shapiro

Corresponding Author(s): B. Jesse Shapiro, McGill University

Review Timeline:

Submission Date:	July 11, 2021
Editorial Decision:	July 26, 2021
Revision Received:	July 30, 2021
Accepted:	July 30, 2021

Editor: Sean Gibbons

Reviewer(s): The reviewers have opted to remain anonymous.

Transaction Report:

DOI: <https://doi.org/10.1128/mSystems.00889-21>

July 26, 2021

Dr. B. Jesse Shapiro
McGill University
Montreal
Canada

Re: mSystems00889-21 (A combination of metagenomic and cultivation approaches reveals hypermutator phenotypes within *Vibrio cholerae* infected patients)

Dear Dr. B. Jesse Shapiro:

Thank you for submitting your manuscript to mSystems. We have completed our review and I am pleased to inform you that, in principle, we expect to accept it for publication in mSystems. However, acceptance will not be final until you have adequately addressed the latest reviewer comments.

Preparing Revision Guidelines

For complete guidelines on revision requirements for your article type, please see the journal Article Types requirement at <https://journals.asm.org/journal/mSystems/article-types>. **Submissions of a paper that does not conform to mSystems guidelines will delay acceptance of your manuscript.**

Sincerely,

Sean Gibbons

Editor, mSystems

Journals Department
Reviewer comments:

Reviewer #2 (Comments for the Author):

The authors generally addressed the reviewers' concerns and I recommend the manuscript be published in mSystems after addressing my remaining concerns. The narrow focus of the revised abstract is a strength. I want to particularly commend the authors for their thoughtful revisions when fielding some of these concerns - the paragraph on iSNVs and co-infection will be helpful for readers.

The paper provides sufficient evidence of its major claim that *V. cholerae* hypermutators can evolve within hosts by sequencing samples from symptomatic patients and asymptomatic household contacts.

I do have some remaining concerns that the authors should address before publication:

-Regarding the statistics of DNA repair mutations, the statistics should be presented for all subjects, not just Patient F (who has the strongest signal of hypermutation). An example calculation would be the probability of identifying ≥ 5 subjects with a mutation in a DNA repair pathway given the number of mutations in each subject.

-Line 165: "It would be unlikely for random sequencing errors to occur in the exact same four sites on two consecutive days by chance alone therefore these iSNVs are likely true positives." I strongly disagree with this statement. Most false positive iSNVs result from mapping error, and thus are reproducible. Interestingly, mapping errors tend to have reproducible frequency-so I'd be more willing to accept this argument if these reproducible iSNVs changed dramatically in frequency across the timepoints.

-Line 587 - 590: "We did not detect any iSNVs among the five isolates sequenced from patient 58.00. In contrast, the metagenomic analysis of patient N revealed seven iSNVs (Table 1), suggesting a potentially higher sensitivity for the detection of rare variants, or possibly false-positive iSNVs inferred from metagenomic reading mapping compared to isolate sequencing."

This could be reworded to state that when only shallow isolate WGS is possible, metagenomics is more appropriate for the detection of iSNVs. Overall, the authors should take another pass through the manuscript for claims that were toned down in response to the first round of reviews.

Line 252: "This suggests that deleterious mutations in hypermutators could be counterbalanced by adaptive mutations that maintain growth." This statement is too strong given that variation in iRep estimates are not well understood and probably driven by noise.

Line 279: "Together, these analyses suggest that *V. cholerae* hypermutators produce NS mutations that are predominantly deleterious or neutral". While this is technically true, it is likely that these hypermutators may have just as many adaptive mutations, but the ability to detect them is drowned out by additional noise.

-Line 337: "Among the other index cases, we found no iSNVs in patient 58.00". Clarify that this is for isolates only.

-mutL mutations lead to an excess of transition (not transversion!) mutations of all types. This is wrong on line 234.

-Figure S2 should have a sense of error, either through plotting of absolute number or with error bars

-It is super confusing to use two schemes to refer to the same patient (e.g. N and 58.00). This gets particularly problematic in the pan-genome section, when I cannot compare isolate pangenomes to the iSNV data.

-Line 342: How does an isolate have mutations? What this is reference to should be stated.

-Supplementary Table 1-Why does each subject have multiple household numbers?

Reviewer #3 (Comments for the Author):

The authors adequately addressed my previous comments in this version of the manuscript. I thank the authors for their careful attention to each of the reviewers' comments.

New minor concern:

Line 192 - 194 - "No iSNVs were observed at the same nucleotide position in different patients, suggesting that iSNVs rarely spread by homologous recombination...". I wonder whether this is sufficient evidence? Perhaps this could benefit from additional explanation?

Reviewer #2 (Comments for the Author):

*The authors generally addressed the reviewers' concerns and I recommend the manuscript be published in mSystems after addressing my remaining concerns. The narrow focus of the revised abstract is a strength. I want to particularly commend the authors for their thoughtful revisions when fielding some of these concerns - the paragraph on iSNVs and co-infection will be helpful for readers. The paper provides sufficient evidence of its major claim that *V. cholerae* hypermutators can evolve within hosts by sequencing samples from symptomatic patients and asymptomatic household contacts. I do have some remaining concerns that the authors should address before publication:*

Response: We thank the reviewer for their positive assessment, and for the previous round of reviews that significantly improved the manuscript. We address the remaining suggestions as detailed below.

-Regarding the statistics of DNA repair mutations, the statistics should be presented for all subjects, not just Patient F (who has the strongest signal of hypermutation). An example calculation would be the probability of identifying ≥ 5 subjects with a mutation in a DNA repair pathway given the number of mutations in each subject.

Response: Thank you for this suggestion. We have added the following text to the beginning of the of the Results section on hypermutators:

“Assuming that DNA repair genes are of average length and contain an average number of NS sites, we can estimate the one-sided binomial probability that NS mutations occur in the observed number of DNA repair genes in each of these five patients (Table 1). We calculated this probability assuming a binomial success rate of 0.0127 (obtained by dividing 51, the number of DNA repair genes (GO:0006281) by 4007, the total number of genes in the *V. cholerae* N16961 reference genome). By multiplying the probabilities from each patient, we obtain an overall probability of 0.0023 that we would see the observed number of DNA repair genes with NS mutations in all five patients. This number of patients with mutated DNA repair genes is therefore unlikely to have occurred by chance alone, given the observed number of mutations.”

-Line 165: "It would be unlikely for random sequencing errors to occur in the exact same four sites on two consecutive days by chance alone therefore these iSNVs are likely true positives." I strongly disagree with this statement. Most false positive iSNVs result from mapping error, and thus are reproducible. Interestingly, mapping errors tend to have reproducible frequency-so I'd be more willing to accept this argument if these reproducible iSNVs changed dramatically in frequency across the timepoints.

Response: We agree that it is difficult to fully exclude the possibility of sequencing or mapping errors here, and we have adjusted the text to reflect this. As suggested, we checked the minor allele frequencies at these four positions at the two sampled time points (with coverage X):

- position 755: 0.055 (163X) and 0.05 (100X)
- position 13163: 0.1 (44X) and 0.25 (26X)
- position 34509: 0.05 (56X) and 0.07 (39X)
- position 53558: 0.4 (30X) and 0.375 (28X)

The frequencies are comparable, except for position 13163, which has relatively low coverage. Therefore, most positions are consistent with the reviewer's hypothesis that mapping errors should

occur at similar frequencies. On the other hand, a systematic mapping error would be expected to occur in other samples, not just the two from the same patient. This is not the case, as we observed no nucleotide positions with iSNVs in more than one patient. We therefore adjusted the text as follows, which we believe succinctly captures the uncertainty:

“It would be unlikely for random sequencing errors to occur in the exact same four sites on two consecutive days by chance alone, therefore these iSNVs are likely either true positives or systematic (site-specific) sequencing or read mapping errors. However, systematic errors would be expected to be seen in other samples at the same nucleotide positions, which is not the case.”

-Line 587 - 590: "We did not detect any iSNVs among the five isolates sequenced from patient 58.00. In contrast, the metagenomic analysis of patient N revealed seven iSNVs (Table 1), suggesting a potentially higher sensitivity for the detection of rare variants, or possibly false-positive iSNVs inferred from metagenomic reading mapping compared to isolate sequencing." This could be reworded to state that when only shallow isolate WGS is possible, metagenomics is more appropriate for the detection of iSNVs. Overall, the authors should take another pass through the manuscript for claims that were toned down in response to the first round of reviews.

Response: We agree with this suggestion, and have reworded this section as follows:

“In contrast, the metagenomic analysis of patient N revealed seven iSNVs (Table 1), suggesting a higher sensitivity for the detection of rare variants which could be easily missed by sequencing only a few isolates. Despite a potentially higher error rate, metagenomics is more appropriate for sensitively detecting iSNVs when only shallow isolate sequencing is possible.”

Line 252: "This suggests that deleterious mutations in hypermutators could be counterbalanced by adaptive mutations that maintain growth." This statement is too strong given that variation in iRep estimates are not well understood and probably driven by noise.

Response: We agree that this statement was too speculative, and we have removed it and replaced it with the following, as suggested:

“This lack of association could be due to noisy replication rate estimates from iRep, and could be revisited in larger patient cohorts.”

Line 279: "Together, these analyses suggest that V. cholerae hypermutators produce NS mutations that are predominantly deleterious or neutral". While this is technically true, it is likely that these hypermutators may have just as many adaptive mutations, but the ability to detect them is drowned out by additional noise.

Response: We agree, and have added the following sentence to clarify this point:

“This does not exclude the possibility of adaptive mutations in hypermutators, but these are difficult to pinpoint against the overwhelming background of non-adaptive mutations.”

-Line 337: "Among the other index cases, we found no iSNVs in patient 58.00". Clarify that this is for isolates only.

Response: We agree and have modified this sentence as follows:

“Among the other index cases, we found no iSNVs in the isolates from patient N”

-mutL mutations lead to an excess of transition (not transversion!) mutations of all types. This is wrong on line 234.

Response: Thank you for catching this error. We have now corrected it as follows:

“For instance, it has been shown in other bacterial pathogens that mutations in *mutT* and *mutL* lead to strong mutator phenotypes, increasing the rate of A:T→C:G transversions and G:C →A:T transitions respectively (34), which we observed in patients (F and I) containing these mutations (Table 1, Fig. S2).”

-Figure S2 should have a sense of error, either through plotting of absolute number or with error bars

Response: Thank you for this suggestion. We found that plotting the absolute number of iSNVs made it difficult to compare the patients, which range from 6 to 207 iSNVs in these plots. To make the panels visually comparable while also showing the absolute numbers, we have now added the number of iSNVs to the header of each panel. We believe this now makes the sampling error clear.

-It is super confusing to use to schemes to refer to the same patient (e.g. N and 58.00). This gets particularly problematic in the pan-genome section, when I cannot compare isolate pangenomes to the iSNV data.

Response: We apologize for this confusion. Patient N (also called 58.00) was the only patient with both a metagenome and isolate genome sequences. For clarity, we now refer to this patient uniquely as Patient N, in both the manuscript text, Figure 3 (which illustrates the pangenome analysis), and Table S1. We believe this is now clear in the following sentence and the paragraph that follows:

“The index case from household 58 (patient N) was the only sample also included in the metagenomic analysis described above, allowing a comparison between culture-dependent and -independent assessments of within-patient diversity.”

-Line 342: How does an isolate have mutations? What this is reference to should be stated.

Response: We have clarified this sentence as follows:

“One isolate sampled from this contact had the highest number of mutations seen in any branch in the phylogeny (five NS mutations, all G:C→T:A transversions) relative to its ancestral branch (*i.e.* to the other isolates from the same person).”

-Supplementary Table 1-Why does each subject have multiple household numbers?

Response: Each row in this table is actually a specific person, but we agree that the lack of ID for some of them makes it unclear. In the update Table S1, we have added a specific ID and the accession number of the reads for each sample.

Reviewer #3 (Comments for the Author):

The authors adequately addressed my previous comments in this version of the manuscript. I thank the authors for their careful attention to each of the reviewers' comments.

Response: We thank the reviewer for their positive assessment, and for the comments that improved the last version of the manuscript.

New minor concern:

Line 192 - 194 - "No iSNVs were observed at the same nucleotide position in different patients, suggesting that iSNVs rarely spread by homologous recombination...". I wonder whether this is sufficient evidence? Perhaps this could benefit from additional explanation?

Response: Thank you for raising this point. Our thinking here was that if iSNVs arose by homologous recombination between different *V. cholerae* strains, the exact same iSNVs would be observed in multiple samples. However, convergent point mutation is an equally plausible explanation, in the absence of any other signal of recombination (e.g. clusters of nearby mutations). Given the absence of evidence, and the lack of a thorough analysis of recombination events, we have chosen to delete this sentence. We now briefly refer to recombination as follows:

“iSNVs were distributed across the genome (Fig. 2A), rather than clustered in hotspots as would be expected if iSNVs arose from recombination events (29). Recombination thus appears to be an unlikely source of iSNVs, although further work would be needed to confirm this.”

July 30, 2021

Dr. B. Jesse Shapiro
McGill University
Montreal
Canada

Re: mSystems00889-21R1 (A combination of metagenomic and cultivation approaches reveals hypermutator phenotypes within *Vibrio cholerae* infected patients)

Dear Dr. B. Jesse Shapiro:

Your manuscript has been accepted, and I am forwarding it to the ASM Journals Department for publication. For your reference, ASM Journals' address is given below. Before it can be scheduled for publication, your manuscript will be checked by the mSystems senior production editor, Ellie Ghatineh, to make sure that all elements meet the technical requirements for publication. She will contact you if anything needs to be revised before copyediting and production can begin. Otherwise, you will be notified when your proofs are ready to be viewed.

As an open-access publication, mSystems receives no financial support from paid subscriptions and depends on authors' prompt payment of publication fees as soon as their articles are accepted. =

Publication Fees:

We recognize that the video files can become quite large, and so to avoid quality loss ASM suggests sending the video file via <https://www.wetransfer.com/>. When you have a final version of the video and the still ready to share, please send it to Ellie Ghatineh at eghatineh@asmusa.org.

Sincerely,

Sean Gibbons
Editor, mSystems

Journals Department
Fig S4: Accept
Table S2: Accept
Table S5: Accept
Fig S2: Accept
Fig S1: Accept
Table S6: Accept
Fig S3: Accept
Table S4: Accept
Table S1: Accept
Table S3: Accept